# Groundwater Storage Changes in the Major North African Transboundary Aquifer Systems during the GRACE Era (2003–2016)

**Frédéric Frappart** 

LEGOS, Université de Toulouse, CNES, CNRS, IRD, UPS—14 Avenue Edouard Belin, 31400 Toulouse, France; frederic.frappart@legos.obs-mip.fr

**Abstract:** Groundwater is an essential component of the terrestrial water cycle and a key resource for supplying water to billions of people and for sustaining domestic and economic (agricultural and industrial) activities, especially in arid and semi-arid areas. The goal of this study is to analyze the recent groundwater changes which occurred in the major North African transboundary aquifers in the beginning of the 21st century. Groundwater storage anomalies were obtained by removing soil moisture in the root zone (and surface water in the case of the Nubian Sandstone Aquifer System) from the terrestrial water storage anomalies estimated using the Gravity Recovery and Climate Experiment (GRACE) over the 2003–2016 time period. Spatio-temporal changes in groundwater storage contrast significantly among the different transboundary aquifers. Low changes (lower than 10 km$^3$) were observed in the Tindouf Aquifer System but they were found to be highly correlated (R = 0.74) to atmospheric fluxes (precipitation minus evapotranspiration, P − ET) at annual scale. The GRACE data revealed huge water loss in the North Western Sahara and the Nubian Sandstone Aquifer Systems, above 30 km$^3$ and around 50 km$^3$, respectively. In the former case, the aquifer depletion can be attributed to both climate (R = 0.67 against P − ET) and water abstraction, and only to water abstraction in the latter case. The increase in water abstraction results from an increase in irrigated areas and population growth. For these two aquifers, a deceleration in the water loss observed after 2013 is likely to be attributed either to an increase in rainfall favoring rain-fed agriculture or to measures taken to reduce the over-exploitation of the groundwater resources.

**Keywords:** groundwater; transboundary aquifers; North Africa; GRACE

## 1. Introduction

The water stored in the aquifers only represents 2.5% of the water present on Earth but around 50% of the freshwater available globally [1]. Aquifer systems are a key component of the global hydrological and the biogeochemical cycles, and also play a crucial role in the sustainability of ecosystems [2–5]. Groundwater resources are essential for the supply of drinkable water to billions of people and sustaining agricultural, industrial, and domestic activities [6,7]. They often constitute the ultimate freshwater resource available to supply water for irrigation and domestic use in semi-arid areas and densely populated countries once a surface water resource is depleted [7]. The groundwater resources have been increasingly used causing their depletion in many regions around the world [8–10]. The excess in groundwater abstraction compared to groundwater recharge, or groundwater depletion, strongly and negatively impacts the environment and the groundwater quality when lasting over long periods [10,11]. The current global overexploitation of groundwater resources exceeds by 3.5 times their recharge [7]. According to Gleeson et al. [9], about 1.7 billion of the world's population lived in regions where the groundwater resources were over-exploited in 2012. Climate change is expected to

intensify the pressure on groundwater resources through a decrease on groundwater recharge and an increase on groundwater use [12–14].

The Gravity Recovery and Climate Experiment (GRACE) mission, launched in 2002, enables to estimate tiny changes of the Earth's gravity field over time. These changes are related to the temporal variations in terrestrial water storage (TWS) [15]. The GRACE mission offered new opportunities for the monitoring of the hydrological cycle [15–17]. TWS is the integrated water content in all storages above and underneath the surface of the Earth (i.e., the sum of surface water, soil moisture, groundwater, and of the water present in the form of snow and ice and contained in the permafrost and biomass). GRACE land water solutions are now commonly used to isolate the groundwater (GW) storage changes from the temporal anomalies of TWS, in combination with external hydrological information from in situ data, remotely-sensed observations and/or model outputs, or through assimilation into hydrological models (see [18] for a recent review). Over arid and semi-arid areas of Africa, GRACE land water solutions were used to reveal the depletion in the North Western Sahara Aquifer System [19–21] and the Nubian Sandstone Aquifer System [22–25], and the increase in GW storage in the Niger River Basin [26] and the Chad Lake Basin [27].

The goal of this study is to determine the importance of climate and anthropogenic factors on the GW changes determined using the GRACE-based TWS in the three major transboundary aquifers located in arid North Africa (i.e., the Tindouf, North Western Sahara and Nubian Sandstone aquifer systems) during the GRACE era (i.e., 2003–2016). The study areas and the datasets used to derive time variations of GW anomalies and hydrological fluxes (rainfall and evapotranspiration) are first presented. The methods are then described. The results consist in the analysis of the consistency of GW storage (GWS) temporal variations and in the determination of the spatio-temporal changes in GW storage in three different transboundary aquifers. The results obtained are discussed taking into account the climate forcing and the anthropogenic effects related to water abstraction.

## 2. Study Area and Datasets

### 2.1. Study Areas

The study area is composed of the three major transboundary aquifer systems (TBA) located in arid North Africa: the Tindouf, North West Sahara and Nubian Sandstone aquifer systems—TAS, NWSAS and NSAS respectively (see Figure 1 for their location).

The TAS is located over Algerian, Moroccan, Western Saharan, and Mauritanian territories, between 5–10° W and 25.4–30° N and covers an approximate area of 180,000 km$^2$ [28]. The TAS is composed of layers of sand from the surface to below 50 m, layers of clay down to 330 m of depth, and below sandstones and quartz [29]. It is located in in a desert climate zone, characterized by an annual air temperature of 24 °C with large seasonal fluctuations (5 °C in January to 45 °C in July). Annual rainfall is average of 50 mm yr$^{-1}$ with large spatial and seasonal variations.

The NWSAS is located over Algerian (69%), Tunisian (8%) and Libyan (23%) territories, between 2.25° W–16.5° E and 26–35° N and covers an approximate area of 1,190,000 km$^2$ [28,30]. NSAS is considered as a multi-layered system of aquifers which embodies a huge stock of non-renewable, fossil water which form two major reservoirs known as the Continental Intercalary (CI) and the Complex Terminal (CT) [31]. It exhibits a porous and fissured/fractured structure. Its depth ranges from around 400 up to 2000 m below the surface. The CI is composed of multiple layers with different lithology: mostly continental sandstone, marine limestone and clay formations. The CT presents has a quite heterogeneous composition which includes carbonated formations of the Upper Cretaceous and detritus episodes of the Tertiary and, mostly, the Miocene [32].

The NSAS is located over Libyan (34.7%), Egyptian (37.5%), Sudanese (17.1%) and Chadian (10.7%) territories, between 17.5–23.6° E and 13–33° N and covers an approximated area ranging from 2,200,000 to 2,600,000 km$^2$ [28,33,34]. It is composed of two parts: the Nubian aquifer, below 26° N, and the Post-Nubian reservoir, located in Libya and Egypt. The NSAS is under confined conditions in

the Nubian aquifer and under unconfined conditions in the Post-Nubian reservoir [33,34]. Its depth varies a lot, ranging from 500 to 4000 m. It is composed of thick sandstone which are sediments from Paleozoic to Permotriassic eras. Limestone and shales from the upper Cretaceous and the Tertiary are intercalated in the north. Saline water fills the sediments in the north and the north-west [35].

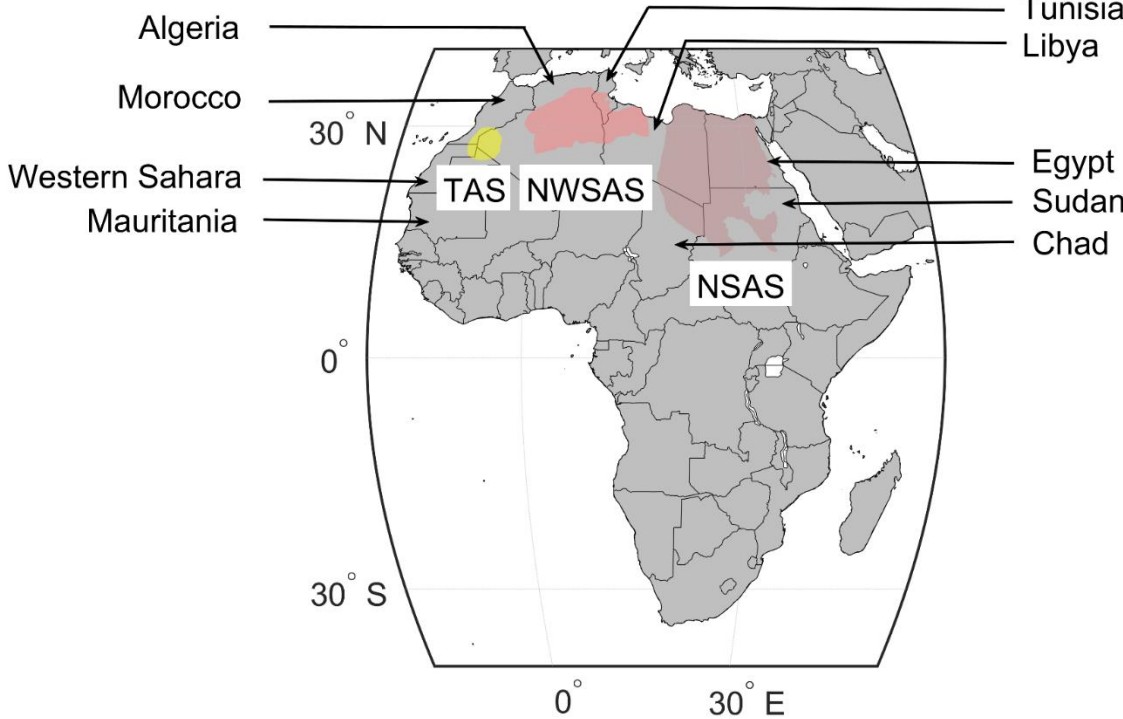

**Figure 1.** Location of the Tindouf Aquifer System (TAS), of the North Western Sahara Aquifer System (NWSAS) and of the Nubian Sandstone Aquifer System (NSAS) in Africa. TBA boundaries were provided by [36]. In this database, the area of these TBA are 221,019, 1,279,963 and 2,892,867 $km^2$ for TAS, NWSAS, and NSAS respectively.

### 2.2. Gravity Recovery and Climate Experiment (GRACE) Mascon-Based Terrestrial Water Storage (TWS)

GRACE mass variations were generally estimated from the changes in the Earth's gravitational potential using the spherical harmonic expansion [15]. Due to model de-aliasing and instrument errors and a lower observability in the east–west direction, the estimates of the Stokes' high-degree harmonic coefficients are impacted by large spurious noise responsible for unrealistic north-south stripes present in the unconstrained GRACE spherical harmonic solutions [15]. To solve this issue, one of the solution was to develop an alternate local approach consisting in estimating mass variations inside mass-concentration (mascon) blocks [37]. Several mascon land water solutions are used in this study.

#### 2.2.1. CSR GRACE RL06 Mascon Solutions

The Cneter for Space Research, University of Texas at Austin (CSR) RL06 Mascon monthly solutions are estimated on a $1° \times 1°$ grid using a Tikhonov regularization constraint only based on the GRACE information [38]. The degree-1 coefficients corrections, corresponding to the geocenter motions between the center of mass of the Earth and the center of figure of the Earth's surface, caused by the longest wavelengths of the mass transport deforming the Earth, are applied. The low degree spherical harmonic coefficient $C_{20}$ is the largest component of the time-variable gravity field of the Earth. However, the $C_{20}$ coefficients from GRACE, strongly impacted by aliasing errors, are replaced with the $C_{20}$ from satellite laser ranging [39]. A glacial isostatic adjustment (GIA) correction, representing the

adjustment process of the Earth to an equilibrium state when loaded by ice sheets, was applied based on the model ICE6G-D [40]. CSR RL06 Mascon monthly solutions, resampled at the spatial resolution 0.25°, are available at [41] from April 2002 to June 2017.

### 2.2.2. JPL GRACE RL06 Mascon Solutions

The Jet Propulsion Laboratory (JPL) RL06M Mascon monthly solutions [42] are estimated on 4551 equal-area 3-degree spherical cap mascons. As for the CSR RL06 solutions, degree-1 coefficients and ICE6G-D GIA [40] corrections are applied and $C_{20}$ coefficients from GRACE are replaced with the $C_{20}$ from satellite laser ranging [39]. JPL RL06 Mascon monthly solutions, resampled at the spatial resolution 0.5°, are available at [43] from April 2002 to June 2017. Two versions of the data are available: with and without the Coastline Resolution Improvement (CRI) filter developed to separate the land and ocean effects for the mascons lying on coastlines [44].

### 2.3. GLEAM

The Global Land Evaporation Amsterdam Model (GLEAM) simulates evaporation and root-zone soil moisture using satellite data [45,46]. Potential evapotranspiration is estimated using observations of surface net radiation and near-surface air temperature based on the Priestley and Taylor equation. It is converted into actual evaporation using a multiplicative, evaporative stress factor. This factor is derived from the vegetation optical depth (VOD), and the root-zone soil moisture (SM) simulations. The VOD parameterizes the extinction effects due to the vegetation affecting the microwave radiations propagating through the vegetation canopy and is related to the vegetation water content [47]. Root zone SM is obtained using a multi-layer soil profile of the rainfall infiltration. Rainfall interception loss is simulated using the Gash model [48]. GLEAM version 3.3 contains two datasets:

- v3.3a which is a global record of actual evaporation and its different component, as well as root zone SM derived from satellite-based SSM, VOD, and snow water equivalent (SWE), reanalysis air temperature and radiation, and a multi-source precipitation product
- v3.3b which is fully satellite-based and quasi-global (50° N–50° S) [46].

These datasets are available upon request at [49] from January 1980 to December 2018 and from June 2003 to December 2018 for v3.3a and v3.3b respectively, at a spatial resolution of 0.25°, and at the daily and monthly temporal resolutions.

### 2.4. Water Volume of Lake Nasser from Hydroweb

The Hydroweb database, developed by Collecte Localisation Satellites/Centre National d'Etudes Spatiales/Laboratoire d'Etudes en Géophysique et Océanographie Spatiales (CLS/CNES/LEGOS) [50], made available several thousands of time series of water levels of rivers and lakes from multi-mission radar altimetry [51]. For a large number of lakes, estimates of inundated surfaces are derived from MODerate resolution Imaging Spectroradiometer (MODIS) multispectral images thresholding spectral bands and indices [52], and combining water level and extent, estimates of water volume assuming a pyramidal shape for the lake bathymetry as in [53–56]. For Lake Nasser, the time series of water level and volume are available from September 1982 to March 2020.

### 2.5. Monthly Rainfall from Tropical Rainfall Measuring Mission/Multi-Satellite Precipitation Analysis (TRMM/TMPA) 3B43

The Tropical Rainfall Measuring Mission/Multi-Satellite Precipitation Analysis (TRMM/TMPA) 3B43 v7 product is a combination of monthly rainfall estimates at a spatial resolution of 0.25°. This dataset is the combination of satellite information from the passive TRMM Microwave Imager (TMI) and Precipitation Radar (PR) onboard the TRMM, from November 1997 to March 2015, the Visible and Infrared Scanner (VIRS) onboard the Special Sensor Microwave Imager (SSM/I) and rain gauge observations. It is currently available from January 1998 to now. The dataset results from the

coupling of the TRMM 3B42-adjusted merged infrared precipitation with the monthly accumulated Climate Assessment Monitoring System or Global Precipitation Climatology Center Rain Gauge analyses [57,58]. It is available on the Goddard Earth Sciences Data and Information Services Center (GES DISC) website [59].

*2.6. European Space Agency (ESA) Climate Change Initiative (CCI) Annual Land-Cover Maps*

The European Space Agency (ESA) Climate Change Initiative (CCI) land-cover (LC) product is composed of LC maps at 300 m of spatial resolution and annual temporal resolution, available from 1992 to 2015. The ESA-CCI LC product was obtained applying the GlobCover unsupervised classification algorithm [60] to daily surface reflectances acquired by the Medium Resolution Imaging Spectrometer (MERIS) at 300 m of spatial resolution between 2003 and 2012 over land surfaces. Additional information acquired by the Advanced Very-High-Resolution Radiometer (AVHRR) at 1 km of spatial resolution between 1992 and 1999, by Système Probatoire d'Observation de la Terre Vegetation (SPOT-VGT) at 1 km of spatial resolution between 1999 and 2013, and by Project for On-Board Autonomy V (PROBA-V) at 100 m of spatial resolution between 2014 and 2015 was used to confirm the classification results and extend the LC annual maps before 2003 and after 2012 through machine-learning techniques [61]. This dataset is available at [62]. These maps provide a description of the Earth's surfaces in 37 LC classes following the United Nations Land Cover Classification System (UN-LCCS) [63]. As discrepancies on cropland areas estimates were reported between the ESA-CCI LC product and Food and Agricultural Organization of the United Nations statistical data (FAOSTAT) in non-Organization for Economic Co-operation and Development (OECD) countries [64], and due to the presence of mosaic pixels including croplands in the study areas, LC classes corresponding to cropland were not included in the analyzes presented below. The only LC class taken into account is the one corresponding to urban areas and their temporal evolution used as an indicator of changes in population.

## 3. Methods

The approach described below was applied. It is composed of the 2 following steps:

1.  As no in-situ dataset of GW level was available over the study areas, the different datasets of TWS and SM used to anomalies of GW storage were compared to determine if they exhibit similar spatio-temporal patterns in order to increase the confidence in the results related to the time variations of GW.
2.  The estimates of temporal anomalies of GW storage as the difference between TWS, SM and surface water storage.

*3.1. Direct Comparisons of the Different Data Sources*

Direct comparisons of the different sources used to compute the anomaly of GW storage were performed. Bias, root mean square difference (*RMSD)*, relative *RMSD* (*RRMSD*) and Pearson correlation coefficient (R) between the different sources of GRACE-based TWS solutions, SM and ET model outputs were estimated at grid-point and basin scales. *RRMSD* was computed as the ratio between the *RMSD* and the average of the mean annual amplitude between years *m* and *n* ($\overline{A}_{m...n}$) of the two compared parameters *a* and *b*:

$$RRMSD(a,b) = \frac{RMSD(a,b)}{\frac{1}{2}\left(\overline{A}_{m...n}(a) + \overline{A}_{m...n}(b)\right)},\tag{1}$$

where $\overline{A}_{m...n}$ is the difference between the mean annual maximum and minimum.

The Mascon GRACE-based TWS products from CSR were downscaled to the lower spatial resolution of the Mascon GRACE-based TWS products from JPL applying a simple average.

*3.2. Groundwater (GW) Storage Anomalies Estimates*

The GRACE mission enables to determine anomalies of Total Water Storage ($\Delta$TWS), i.e., the sum of the anomalies water content of the different hydrological reservoirs present in the study area, e.g., surface water ($\Delta$SW), soil moisture ($\Delta$SM), groundwater ($\Delta$GW), ... :

$$\Delta \text{TWS} = \Delta \text{SW} + \Delta \text{SM} + \Delta \text{GW}, \tag{2}$$

GW storage anomalies were estimated using the commonly used straightforward approach (see e.g., [18] for a recent review) consisting in removing the different hydrological contributions from the other reservoirs to the anomaly of TWS, to isolate the anomaly of GW:

$$\Delta \text{GW} = \Delta \text{TWS} - \Delta \text{SW} - \Delta \text{SM}, \tag{3}$$

Anomalies of SM and SW were obtained removing the mean of these quantities over the whole observation period to their temporal variations.

The study areas are located in arid and semi-arid areas where SW can be neglected. Nevertheless, the NSAS encompasses a part of the Nile Basin. For SW, the contribution of the water stored in the Nile River was neglected following [65], but not water volume variations of the Nasser Lake. The water volume variations from Hydroweb are missing for a couple of months due to missing images that do not allow surface estimates in July 2003 and 2004, from May to August 2005, 2006, 2010 and 2012 and from June to August 2011. To complete these missing values, a third order polynomial relationship was determined between water levels and water volume anomalies (Figure A1). A $R^2$ of 1 and a RMSE of $2.3 \times 10^{-3}$ km$^3$ was found between observations and estimates.

For SM, conversion from volumetric water content (m$^3$ m$^{-3}$) to soil water content (mm) was performed as follows:

$$\Delta \text{SM (mm)} = h_0 \times \Delta \text{SM (m}^3 \text{ m}^{-3}), \tag{4}$$

where $h_0$ is the soil depth (mm).

For GLEAM products, the soil depth over bare soil is 50 mm [45]. Based on the ESA-CCI LC, the land cover is mostly composed of bare areas for the three TBAs between 2002 and 2015. The percentages of bare areas represent around 97 %, for the TAS, around 84% for the NWSAS, and above 79% for the NSAS.

Time-series in the SM and GW reservoirs were computed as follows:

- for the mean value at basin-scale [66]:

$$X(t) = \frac{R_e{}^2}{S} \sum\nolimits_{j \in S} x\big(\lambda_j, \varphi_j, t\big) cos\big(\varphi_j\big) \Delta\lambda \Delta\varphi, \tag{5}$$

where $x$ is $\Delta$TWS, $\Delta$SM or $\Delta$GW at coordinates ($\lambda_j$, $\varphi_j$) and time $t$, $X$ the corresponding basin-scale average of $x$, $R_e$ is the Earth's radius and equals 6378 km, $S$ is the surface of the basin, $\Delta\lambda$ and $\Delta\varphi$ are the grid steps in longitude and latitude, respectively.

- for the corresponding volume $V$ [67]:

$$V_x(t) = R_e{}^2 \sum\nolimits_{j \in S} x\big(\lambda_j, \varphi_j\big) cos\big(\varphi_j\big) \Delta\lambda \Delta\varphi, \tag{6}$$

## 4. Results

*4.1. Direct Comparisons of the Different Datasets*

Due to the lack of in situ data to validate the results that are presented in this study, the first step is to determine if the different datasets used to analyze the spatio-temporal variations of the hydrological

fluxes and storages exhibit a similar behavior. As explained in Section 2.2, GRACE mascon solutions were selected for TWS as they are less affected by spurious meridian noise compared with the global ones. For SM and ET, two versions of GLEAM model outputs were chosen, as this model provides reliable and accurate estimates of these parameters over Africa and semi-arid areas [68–70]. Owing to the use of a medium resolution (250 m) land-cover fraction, larger fractions of bare soil are taken into account in arid and semi-arid areas compared to previous versions and other land-surface models. This leads to an increase in bare soil evaporation and a reduction of vegetation transpiration and a better agreement with in situ data [46]. For surface water, many studies already demonstrated the good accuracy of lake volume variations from the combination of satellite images with radar altimetry (see [71]).

The time series of TWS from CSR, JL CRI and JPL RL06 mascon solutions, and of SM from GLEAM v3.3a and 3.3b over 2003–2016 for TAS, NWSAS and NSAS are presented in Figure 2. The annual amplitude of TWS is rather small (below 20–30 mm), but higher for CSR than for JPL CRI and JPL mascon solutions, and no clear seasonal cycle can be observed (Figure 2a,c,e). Biases and RMSD are found to be below 3 mm and 10.5 mm respectively for all three TBA (Table 1). These latter values correspond to RRMSD ranging from 30 to 45% when comparing TWS from CSR and JPL. R greater than 0.8 were estimated between TWS from CSR and the two JPL solutions for NWSAS and NSAS but only greater or equal 0.5 for TAS (Table 1).

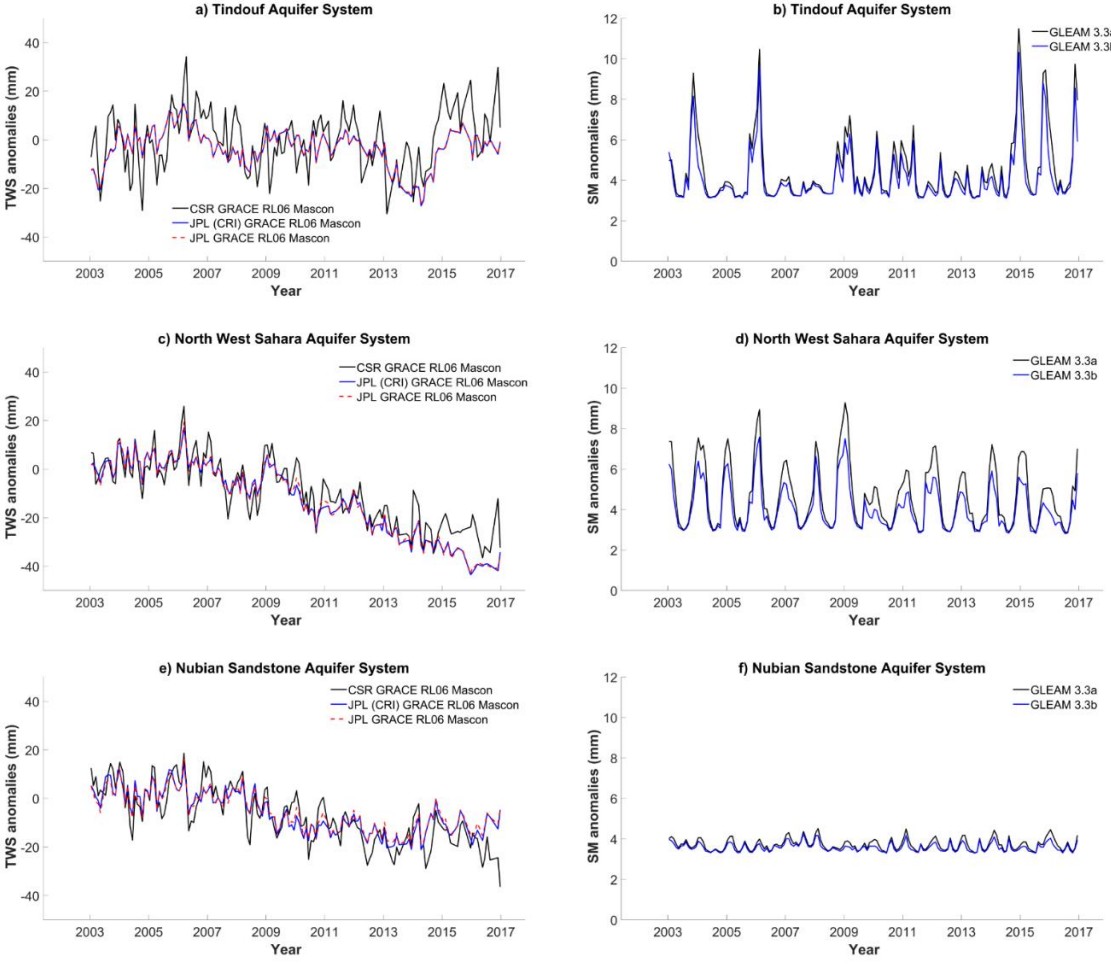

**Figure 2.** Time series of GRACE-based anomalies of TWS from the CSR RL06 (black), JPL (CRI) RL06 (blue), JPL (RL06) (dashed red) Mascon solutions (left panel) and SM anomalies from GLEAM 3.3a (black) and 3.3b (blue) (right panel) for TAS (**a,b**) NWSAS (**c,d**) and NSAS (**e,f**) respectively between 2003 and 2016.

**Table 1.** Statistical results (bias, RMSD, RRMSD and R) of the comparison between the different products used for soil moisture (SM) (Global Land Evaporation Amsterdam Model (GLEAM) 3.3a and 3.3b) and RL06 mascon Gravity Recovery and Climate Experiment (GRACE)-based terrestrial water storage (TWS) (CSR, JPL, JPL Coastline Resolution Improvement (CRI)) in the TAS, NWSAS and NSAS between January 2003 and December 2016.

| Aquifer System | Parameter | Data Sources | Bias (mm) | Bias (km³) | RMSD (mm) | RMSD (km³) | RRMSD (%) | R |
|---|---|---|---|---|---|---|---|---|
| TAS | SM | GLEAM3.3a/3.3b | 0.37 | 0.07 | 0.45 | 0.08 | 11.7 | 1.00 |
| TAS | TWS | CSR/JPL CRI | 2.79 | 0.53 | 10.41 | 1.92 | 45.2 | 0.50 |
| TAS | TWS | CSR/JPL | 2.77 | 0.62 | 10.35 | 1.91 | 45.0 | 0.51 |
| TAS | TWS | JPL CRI/JPL | 0.02 | 0.00 | 0.26 | 0.05 | 1.8 | 0.99 |
| NWSAS | SM | GLEAM3.3a/3.3b | 0.59 | 0.61 | 0.49 | 0.52 | 13.6 | 1.00 |
| NWSAS | TWS | CSR/JPL CRI | 2.56 | 2.72 | 6.30 | 6.63 | 34.0 | 0.91 |
| NWSAS | TWS | CSR/JPL | 2.46 | 2.61 | 6.01 | 6.32 | 31.8 | 0.91 |
| NWSAS | TWS | JPL CRI/JPL | 0.11 | 0.11 | 0.98 | 1.03 | 6.7 | 1.00 |
| NSAS | SM | GLEAM3.3a/3.3b | 0.13 | 0.33 | 0.10 | 0.25 | 13.9 | 1.00 |
| NSAS | TWS | CSR/JPL CRI | −0.80 | −1.94 | 6.51 | 16.01 | 36.2 | 0.83 |
| NSAS | TWS | CSR/JPL | −1.17 | −2.87 | 6.33 | 15.57 | 34.9 | 0.85 |
| NSAS | TWS | JPL CRI/JPL | 0.38 | 0.93 | 1.30 | 3.22 | 9.2 | 0.99 |

The annual variations of SM from GLEAM v3.3a and 3.3b are also rather low (below 10 mm). They exhibit a well-marked seasonal cycle, with a peak in summer, over TAS and NWSAS (Figure 2b,d respectively) but almost no variations over NSAS (Figure 2f). Low biases (below 0.6 mm), low RMSD (below 0.5 mm) corresponding to RRMSD lower than 15% and very high correlations (R = 1) were found over TAS, NWSAS and NSAS for the two versions of GLEAM SSM.

The spatial distribution of the bias, RMSD, RRMSD and R are presented for the comparison between TWS from CSR and JPL CRI, CSR and JPL, JPL CRI and JPL mascon solutions on Figures 3, A2 and A3 respectively, for the comparison between SM from GLEAM v3.3a and v3.3b on Figure 4. Lower agreement is found on the north east part of the TAS, on the north west of the NWSAS and on the north, south and in the Lake Nasser region for the NSAS when comparing TWS from CSR and JPL CRI/JPL solutions (Figures 3 and A2). Very good agreement is found when comparing TWS from JPL CRI and JPL solutions except over coastal areas of the Mediterranean Sea and Red Sea (Figure A3).

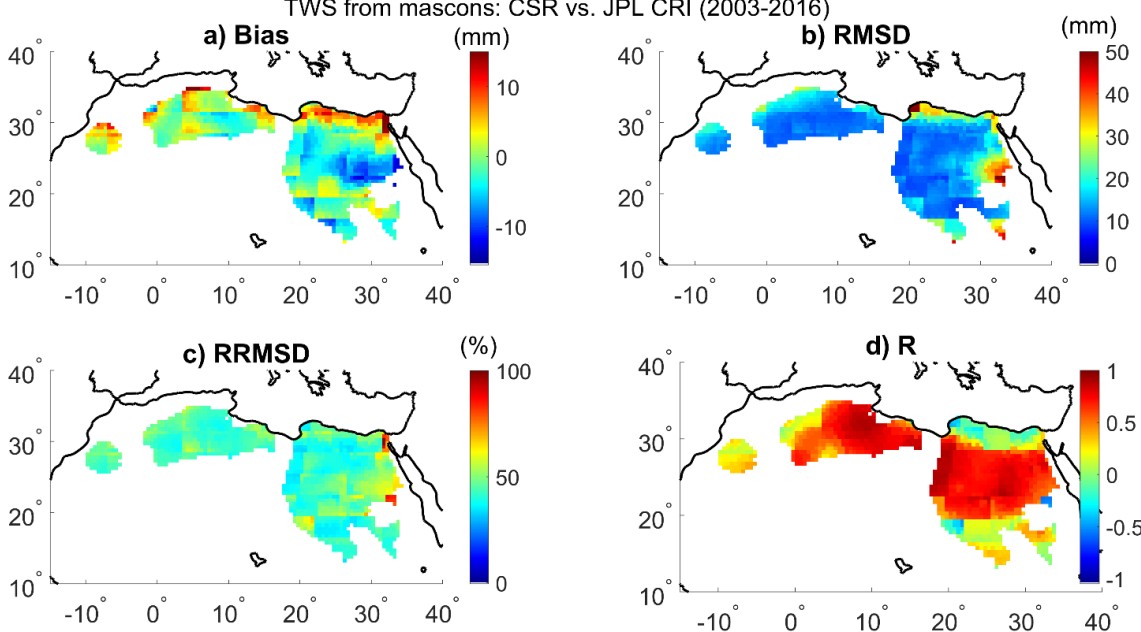

**Figure 3.** Maps of bias (**a**); RMSD (**b**); RRMSD (**c**); R (**d**) between TWS from CSR and JPL CRI RL06 mascon solutions over 2003–2016.

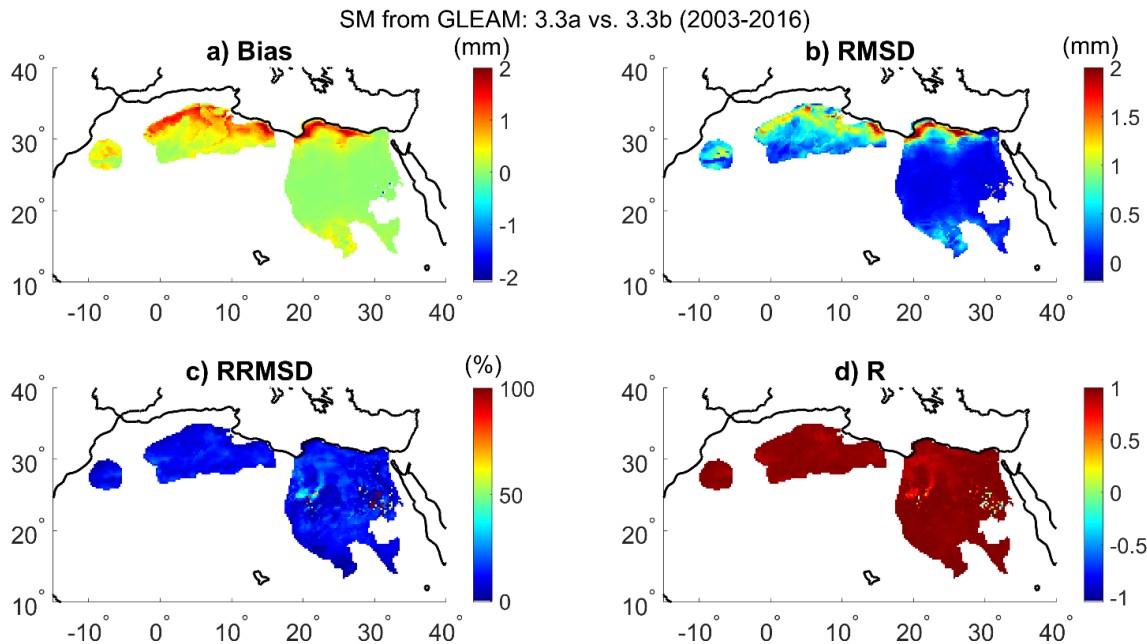

**Figure 4.** Maps of bias (**a**); RMSD (**b**); RRMSD (**c**); R (**d**) between SM from GLEAM v3.3a and v3.3b over 2003–2016.

Very good agreement is also found between the two versions of SM from GLEAM. Larger bias and RMSD are found in the central part of TAS and northern parts of NWSAS and NSAS (Figure 4a,b) and in the northwest and in the northeast (close to Nasser Lake) parts of NSAS (Figure 4c,d). As JPL CRI and JL GRACE-based mascon solutions and GLEAM v3.3a and 3.3b products are very similar, the results will be presented mostly using CSR and JPL CRI mascon solutions and GLEAM v3.3b products in the followings.

### 4.2. Groundwater Storage Spatio-Temporal Variations

Temporal variations of GW storage anomalies, expressed in km$^3$, between 2003 and 2016, along with associated interannual trends obtained using a 13-month averaging window, are presented in Figure 5 over TAS, NWSAS and NSAS. As GW storage anomalies are mostly driven by the GRACE-based TWS, no clear seasonal signal can be observed and higher annual variations can be observed on CSR rather than on JPL CRI and JPL (not shown) solutions. For the GW storage anomalies, the results are similar to those obtained in Table 1 for TWS. When considering the trend on this parameter, a much better agreement is observed with a decrease of the bias and RMSD—this latter estimator is almost divided by two for the NSAS—and an increase of R around 0.1 in each TBA (Table 2).

**Table 2.** Statistical results (bias, RMSD and R) of the comparison between the different groundwater (GW) volume anomalies using SM GLEAM 3.3b and RL06 mascon GRACE-based TWS (CSR and JPL CRI) (and lake volume for NSAS) in the TAS, NWSAS and NSAS between January 2003 and December 2016 for the volume anomalies and July 2003–June 2016 for the associated tends.

| Aquifer System | Parameter | Bias (km$^3$) | RMSD (km$^3$) | R |
|---|---|---|---|---|
| TAS | GW | 0.06 | 1.76 | 0.51 |
| TAS | GW trend | 0.04 | 1.10 | 0.61 |
| NWSAS | GW | 0.61 | 6.25 | 0.92 |
| NWSAS | GW trend | 0.48 | 4.50 | 0.98 |
| NSAS | GW | −0.41 | 14.43 | 0.84 |
| NSAS | GW trend | −0.18 | 7.8 | 0.95 |

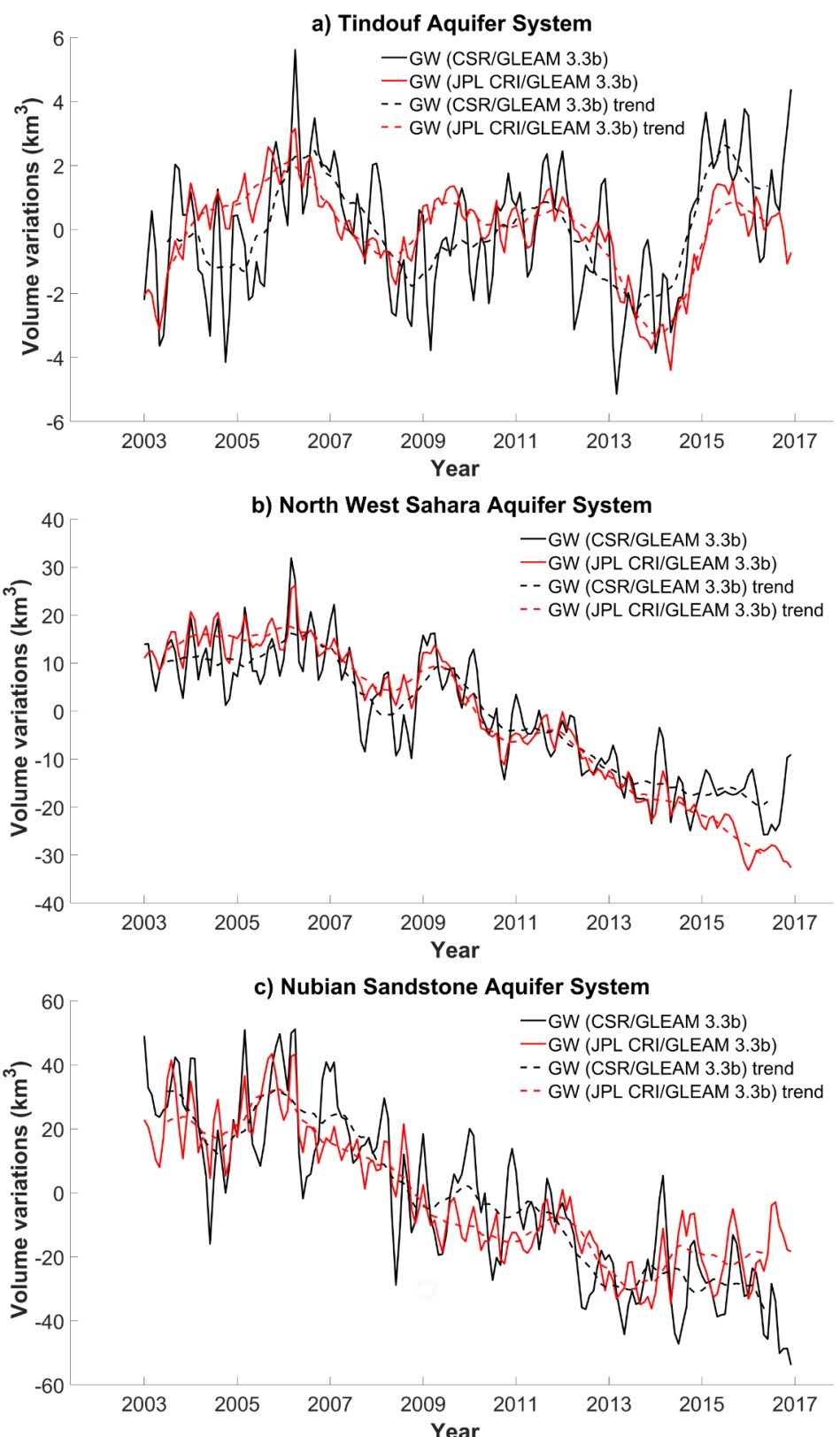

**Figure 5.** Time series of monthly anomalies of GWS derived from GRACE-based TWS from mascon CSR RL06, SM from GLEAM 3.3b (black), from mascon JPL CRI RL06, SM from GLEAM 3.3b (red) and associated trends (dotted lines of the same color) between 2003 and 2016 for TAS (**a**); NWSAS (**b**); and NSAS (**c**).

An interannual signal with a period of 4 years can be observed for the TAS and NWSAS (Figure 5a,b) as well as a decrease in GW storage starting in 2006 for the NWSAS and NSAS, with a deceleration after 2014 (Figure 5b,c). Annual rainfall, evapotranspiration and their difference also exhibit a similar behavior as the temporal variations in GW storage in each TAS (Figure 6).

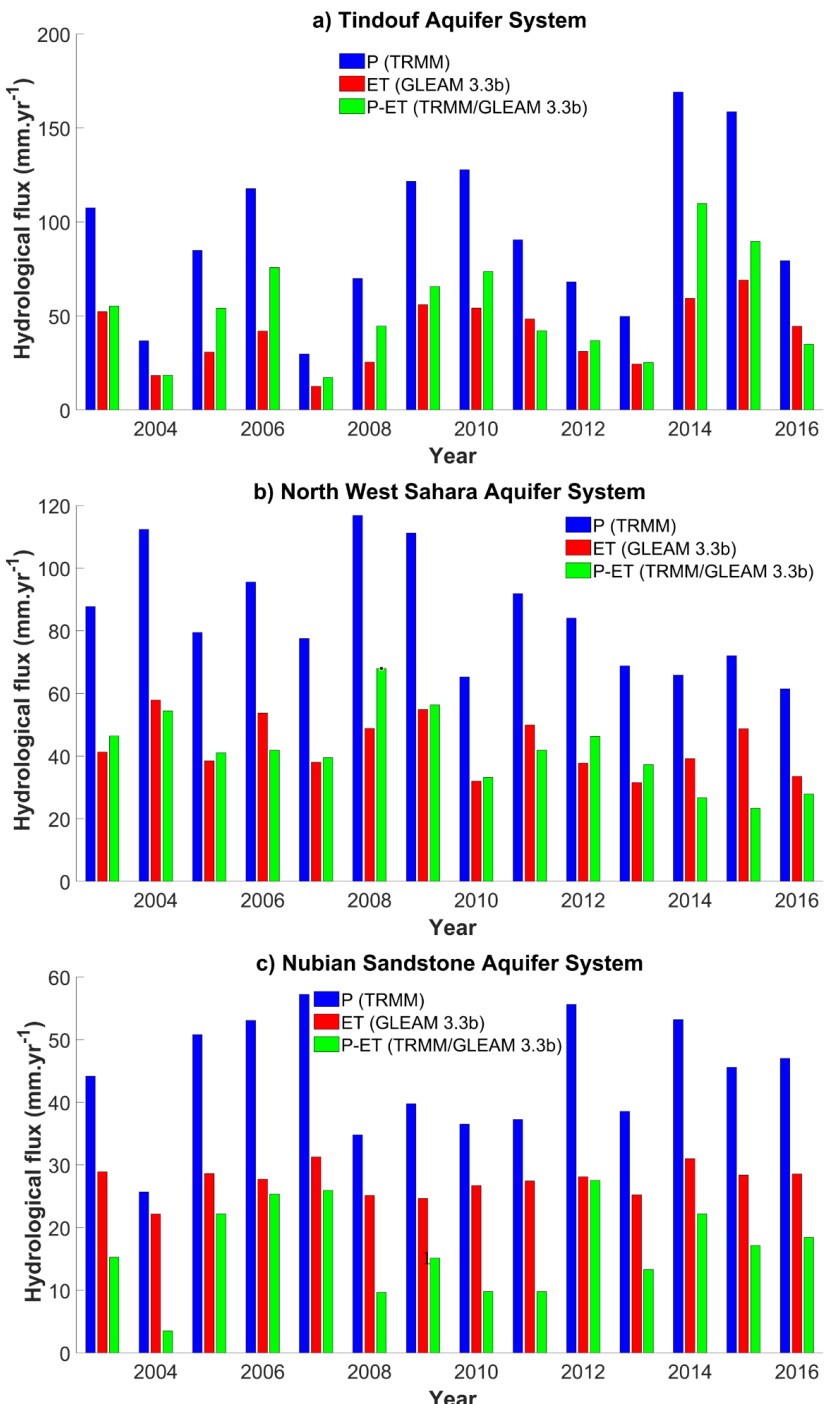

**Figure 6.** Time series of annual rainfall from Tropical Rainfall Measuring Mission (TRMM, blue), evapotranspiration (ET) from GLEAM 3.3b (red), precipitation minus evapotranspiration (P − ET) as the difference of the two former (green) between 2003 and 2016 for TAS (**a**); NWSAS (**b**); and NSAS (**c**).

Maps of GW changes over 2003–2016 were obtained also applying a 13-month averaging window to the grids of groundwater anomalies determined using (3). These changes correspond to the period

ranging from July 2003 to June 2016 (Figure 7). Very similar spatial patterns are observed using either CSR or JPL CRI mascon solutions in combination with either GLEAM 3.3a or 3.3b. More details are present when using the CSR solutions (Figure 7a,c) than using the JPL solutions (Figure 7b,d) as CSR solutions have a higher spatial resolution (0.25° instead of 0.5° for JPL solutions). Low changes are observed over the TAS except a small increase in the north, around 20–30 mm, between 2003 and 2016. By contrast, NWSAS and NSAS exhibit very strong spatial heterogeneities: GW did not vary or increase by a low amount in the western part of the NSWAS between 2003 and 2016 whereas a strong decrease (from −40 to −75 mm) is observed in the eastern part during the same period. GW also strongly decreased in the central part NSAS (from −40 to −90 mm) between 2003 and 2016 and slightly increased (up 30–40 mm) in the northern and the southern parts.

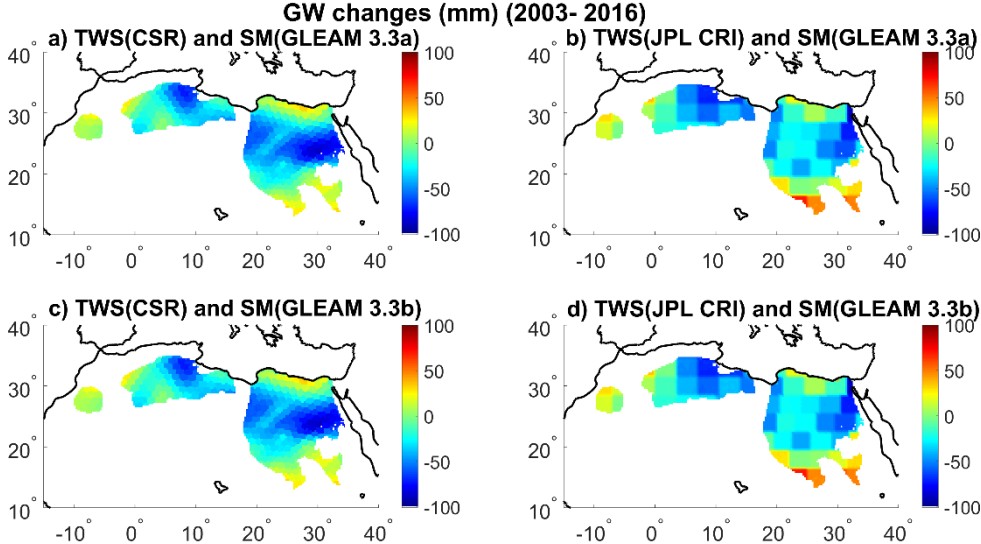

**Figure 7.** Changes in GW storage, expressed in terms of equivalent water height (mm) using GRACE-based TWS from mascon CSR RL06 and SM from GLEAM 3.3a (**a**); GRACE-based TWS from mascon JPL CRI RL06 and SM from GLEAM 3.3a (**b**); GRACE-based TWS from mascon CSR RL06 and SM from GLEAM 3.3b (**c**); and mascon JPL CRI RL06 and SM from GLEAM 3.3b (**d**) between 2003 and 2016 for TAS, NWSAS and NSAS. These changes in GW storage anomaly were computed using a 13-month averaging window.

The interannual variations of the annual P, ET and P − ET were computed using the same 13-month averaging window over 2003–2016 (Figure 8). An opposite pattern is observed over the TAS with an increase in P and P − ET in the south, up to 100–150 mm. The changes in GW exhibits a meridian distribution in the NWSAS, with a decrease down to −100 mm in the north and low variations in the south for P, ET and P − ET. The interannual changes in the NSAS occur in the north with an increase in P, ET and P − ET above 50 mm and in the south with a decrease between 15° and 20° and an increase between 10° and 15° of latitude for P and P − ET and the opposite for ET.

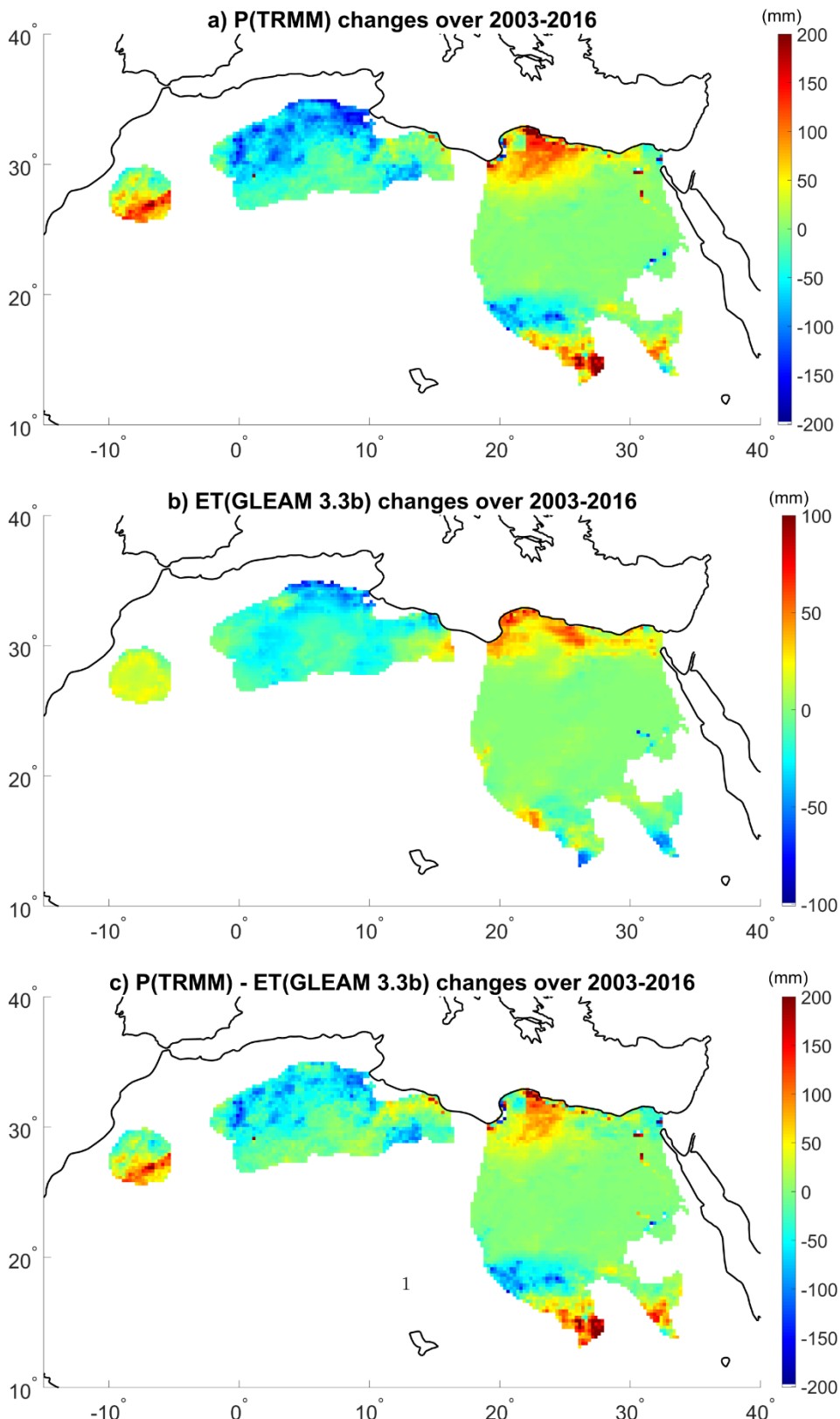

**Figure 8.** Time series of annual rainfall from TRMM (blue), ET from GLEAM 3.3b (red), P–ET as the difference of the two former (green) between 2003 and 2016 for TAS (**a**); NWSAS (**b**); and NSAS (**c**).

## 5. Discussion

Changes in groundwater storage (GWS) can be attributed to both climatic and anthropogenic factors. In the following, the importance of these two factors will be discussed using hydrological fluxes estimates on the one hand, and abstraction values, changes in irrigated surfaces, population, and urbanization on the other hand, when available.

### 5.1. Climate Impacts on Groundwater Storage (GWS)

Considering basin-scale temporal changes, mean annual values of GWS were compared to annual hydrological fluxes (P, ET, P − ET) using cross-correlation (Table 3). In the three TBA, higher correlations were obtained between GWS and P–ET than between GWS and P, and especially than between GWS and ET, the former quantity representing the net contribution of the atmosphere to the land water cycle. High temporal correlation was found between mean annual GWS and P − ET in the TAS using TWS from the CSR mascon products over 2003–2016. The low correlation obtained using the JPL-CRI mascon solutions is likely to be attributed to the lower spatial resolution of this dataset which is not sufficient to provide a reliable monitoring of the TWS, and, hence, of the GWS in a TBA with an area lower than 200,000 $km^2$, and variations in GWS lower than 10 $km^3$ (i.e., lower than 55 mm). In this TBA, the temporal evolution of GWS is strongly linked to the one of P − ET, that is to say to climate factors. The high correlation value suggests that the TAS is not a fossil aquifer. A time-lag of one year was found for the maximum of correlation between the annual forcing in P − ET and GWS. This time-lag is likely to provide an integrated information on the large-scale rainfall-recharge complexity.

**Table 3.** Maximum of cross-correlation and time-lag (ΔT) between the annual average GW volume anomalies obtained using GLEAM 3.3b for SM and RL06 mascon GRACE-based TWS (CSR and JPL CRI) (and lake volume for NSAS) and the hydrological fluxes (P from TRMM 3B43, ET from GLEAM 3.3b and P–ET) in the TAS, NWSAS and NSAS over 2003–2016.

| Aquifer System | Parameter | R (ΔT (Year)) | | | | | |
| --- | --- | --- | --- | --- | --- | --- | --- |
| | | P | | ET | | P − ET | |
| | | CSR | JPL-CRI | CSR | JPL-CRI | CSR | JPL-CRI |
| TAS | GW | 0.65 * (+1) | 0.29 *** (+1) | 0.46 ** (+1) | 0.21 *** (+1) | 0.73 * (+1) | 0.32 *** (+1) |
| TAS | GW smoothed | 0.66 * (+1) | 0.28 *** (+1) | 0.48 ** (+1) | 0.21 *** (+1) | 0.74 * (+1) | 0.31 *** (+1) |
| NWSAS | GW | 0.60 * (0/+1) | 0.66 * (0) | 0.47 ** (0) | 0.47 ** (0) | 0.60 * (0/+1) | 0.66 * (0) |
| NWSAS | GW smoothed | 0.60 * (0/+1) | 0.67 * (0) | 0.48 ** (+1) | 0.46 ** (0) | 0.60 * (0/+1) | 0.67 * (0) |
| NSAS | GW | 0.07 • (0) | 0.07 • (0) | 0.00 • (0) | 0.00 • (0) | 0.09 • (0) | 0.09 • (0) |
| NSAS | GW smoothed | 0.04 • (0) | 0.04 • (0) | −0.03 • (0) | −0.03 • (0) | 0.06 • (0) | 0.06 • (0) |

* $p < 0.05$, ** $p < 0.1$, *** $p < 0.5$, • $p > 0.5$.

A correlation above 0.6 was found between P and P–ET and GWS for the NWSAS with no time-lag (similar correlation values were found for a time-lag of one year) confirming that the NWSAS is not a fossil aquifer, and the assumption from [19,21] on the rainfall–recharge relationship. A water loss of 29.2/42.8 $km^3$ was estimated between 2003 and 2016 using the CSR and the JPL CRI mascon solutions, respectively. The difference between the two estimates mostly comes from the period 2013–2016 where the water loss is 6.9 $km^3$ using the CSR mascon solutions and 16.9 $km^3$ using the JPL CRI mascon solutions. This large discrepancy could be attributed either to the inclusion of signals from a surrounding region due to the coarser resolution of the JPL CRI mascon solutions or to a different impact on the solutions from CSR and JPL of the decrease in altitude of the GRACE satellites. This huge depletion of NWSAS observed after 2009 is not likely to be only attributable to climatic factors as R between GWS and either P or P − ET is below 0.7, even if a large decrease in P and P − ET is observed between 2003 and 2016 with P − ET ranging from 40 to 70 mm yr$^{-1}$ between 2003 and 2009, from 35 to 45 mm yr$^{-1}$ between 2010 and 2013, and around 25 mm yr$^{-1}$ between 2014 and 2016 (Figure 6b).

No correlation was found between the atmospheric hydrological fluxes and the GWS in the NSAS as expected for a fossil aquifer. The NSAS experienced a loss of 54.6/47.6 $km^3$ between 2003 and 2016 using the CSR and the JPL CRI mascon solutions respectively with an acceleration between 2006 and

2013 (Figure 5c). The major decrease in GWS occurred in the center of NSAS, between 20° and 30°, where small increases can be observed below and above this latitude range (Figure 7). Correspondences with the spatial pattern of the hydrometeorological fluxes can be observed with no changes in center of NSAS but an increase at higher and lower latitudes (Figure 8).

*5.2. Anthropogenic Impacts on GWS*

Pumping rates data are difficult to access in these three TBA. To quantify the anthropogenic effect on the temporal evolution of GWS in each of them, different sources were considered: estimates of annual abstraction and population when available, and changes in urban areas during the GRACE era from the ESA-CCI LC at 300 m of spatial resolution.

The use of wells has been declining in the TAS since 1950 [72]. Based on the ESA-CCI LC, urban areas slowly increased from 73 to 78 $km^2$ between 2003 and 2015. A small increase in population over this area can hence be expected. From these different pieces of information, the impact on human activities is most likely negligible in the TAS over 2003–2016.

In the NWSAS, the annual abstraction increased from 2.18 $km^3$ in 2000, to 2.50 $km^3$ in 2008, 3.00 $km^3$ in 2012, and 3.20 in 2018 [73,74]. If we consider these rates of annual water abstraction, they represent almost half of the decrease in GWS over this period (between 10 and 12 $km^3$ for a total loss of 24.1/25.5 (CSR/JPL CRI) $km^3$ over 2008–2012 based on the estimates presented in Figure 5b). The highest depletion rates cover a large part of the Complex Terminal and the south western part of the Continental Intercalaire (Figure 7). These areas correspond to the locations of the wells in the NWSAS (see [73]). Between 2000 and 2020, the irrigated surfaces increased from 2500 (1700, 400, and 400) to 4320 (3200, 550, and 770) $km^2$ in the NWSAS (in Algeria, Tunisia, and Libya respectively), whereas the population increased from 4,800,000 (2,600,000, 1,000,000, and 1,200,000) to 7,000,000 (3,700,000, 1,500,000, and 1,800,000) in the NWSAS (in Algeria, Tunisia, and Libya respectively) over the same period [75]. This increase in population is accompanied by an increase in urban areas from 1073 in 2002 to 1658 $km^2$ in 2015 according to the ESA-CCI LC. If abstraction reaches 7.77 $km^3$ $yr^{-1}$ in 2050 as estimated by [73], due to an increase in irrigated areas and population, estimated to 5130 $km^2$ and 8,800,000 [75], the exploitation of the resource will be highly unstainable

The decrease of GWS observed in the center of NSAS can be attributed to an increase in pumping rather than the evaporation in oases as this latter parameter was found almost constant over the study area (Figure 8). In the early 2000s, withdrawal rates in the whole NSAS reached 2.177 $km^3$ $yr^{-1}$ [76]. Considering a constant abstraction, this withdrawal rate will lead to a decrease in GWS of 30.5 $km^3$ between 2003 and 2016 when a water a loss of 54.6/47.6 (CSR/JPL CRI) $km^3$ was determined. This difference can be accounted for by an increase in pumping that was observed in different locations of NSAS during the last few years in response to huge increases in irrigated areas [77–79], and population (the urban areas vary from 2515 to 4504 $km^2$ over 2002–2015 according to the ESA-CCI LC). Nevertheless, GWS stopped declining from 2013 to 2016 (Figure 5c). Is this due to the increase in rainfall observed from in 2012 and from 2014 to 2016 compared with the previous drier yeas (2008–2011) which could have allowed more rain-fed agriculture or is it the result of a better cooperation between Egypt, Libya, Sudan and Chad to limit the over-exploitation of the aquifer [80]?

**6. Conclusions**

This study provides quantification of the GWS changes over three major TBA located in arid and semi-arid North Africa: TAS, NWSAS and NSAS during the GRACE era (2003–2016). GWS variations were estimated removing SM (and SW storage from for NSAS) from GRACE-based TWS. Mascon solutions from CSR and JPL RL06 were chosen instead of global ones to minimize the effects of leakage. SM were obtained from GLEAM 3.3a and 3.3b outputs, and surface water storage from a combination of radar altimetry and satellite images made available by the Hydroweb database. The different products agreed well with each other in the different TBA, except over TAS for the GRACE solutions (R = 0.5), and their combination also. Nevertheless, JPL mascon solutions suffered from the limited

spatial resolution over TAS, which has a smaller area (180,000 km$^2$). Interannual variations of GWS were estimated using a 13-month averaging window. Interannual variations of GWS in TAS were dominated by an oscillation with a 4-year period also present in P and ET. A correlation of 0.74 was obtained between GWS (using the CSR mascon solutions) and P − ET. For NWSAS, a similar oscillation was observed with a lower correlation between GWS and P − ET (R = 0.6 using CSR mascon solutions and R = 0.67 using JPL mascon solutions) suggesting that NWSAS is not a fossil aquifer. GWS declined during the whole period with an increasing trend between 2009 and 2013 due to both a decrease in P and P − ET and a growth of the withdrawals which represented 40% to 50% of the loss. A large water loss, greater than 45 km$^3$, occurred in NSAS between 2006 and 2013. It was larger than the latest withdrawal rate provided in the early 2000s which led to a decrease in GWS of 30.5 km$^3$. The loss determined using GRACE-based TWS provided evidence of an increase of pumping in NSAS observed locally. For these two aquifers, the increase in abstraction was due to the increase in irrigated areas and population growth.

For NSAS and possibly for NWSAS, the GWS seemed to be quasi-stable between 2013 and 2016. Is it still the case? Is the GWS of TAS still strongly related to hydroclimatic forcings? The launch of the GRACE Follow-On mission in May 2018 allows us to continue monitoring the GW changes of the major aquifers around the world.

**Funding:** This research was funded by the Hydroweb CNES TOSCA grant.

**Acknowledgments:** F.F. wants to express his gratitude to Philippe Guiochon and Conrad Kilian for inspiring this study. We thank two anonymous Reviewers who helped improve the manuscript.

**Conflicts of Interest:** The authors declare no conflict of interest. The funders had no role in the design of the study; in the collection, analyses, or interpretation of data; in the writing of the manuscript; or in the decision to publish the results.

## Appendix A

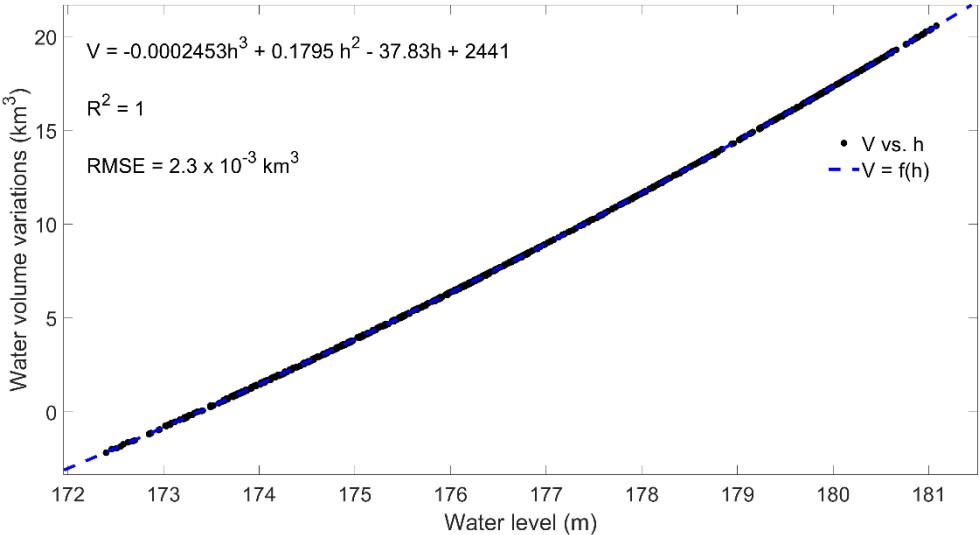

**Figure A1.** Third-order polynomial fit between altimetry-based water level and water volume variations.

## Appendix B

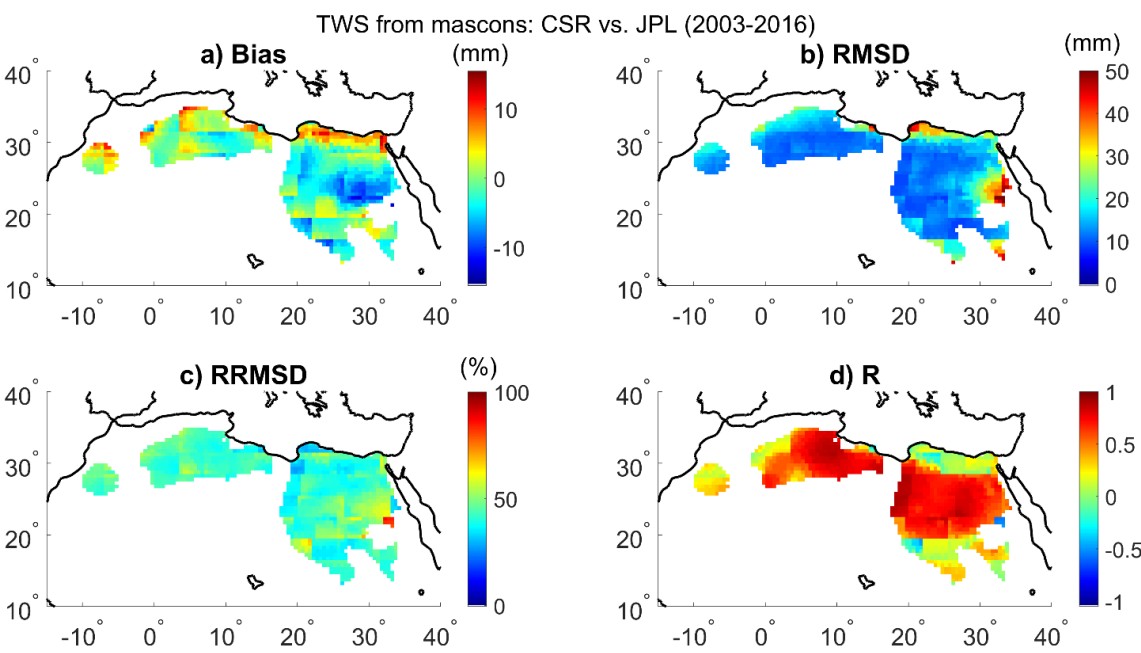

**Figure A2.** Maps of bias (**a**); RMSD (**b**) RRMSD (**c**); R (**d**) between TWS from CSR and JPL mascon solutions over 2003–2016.

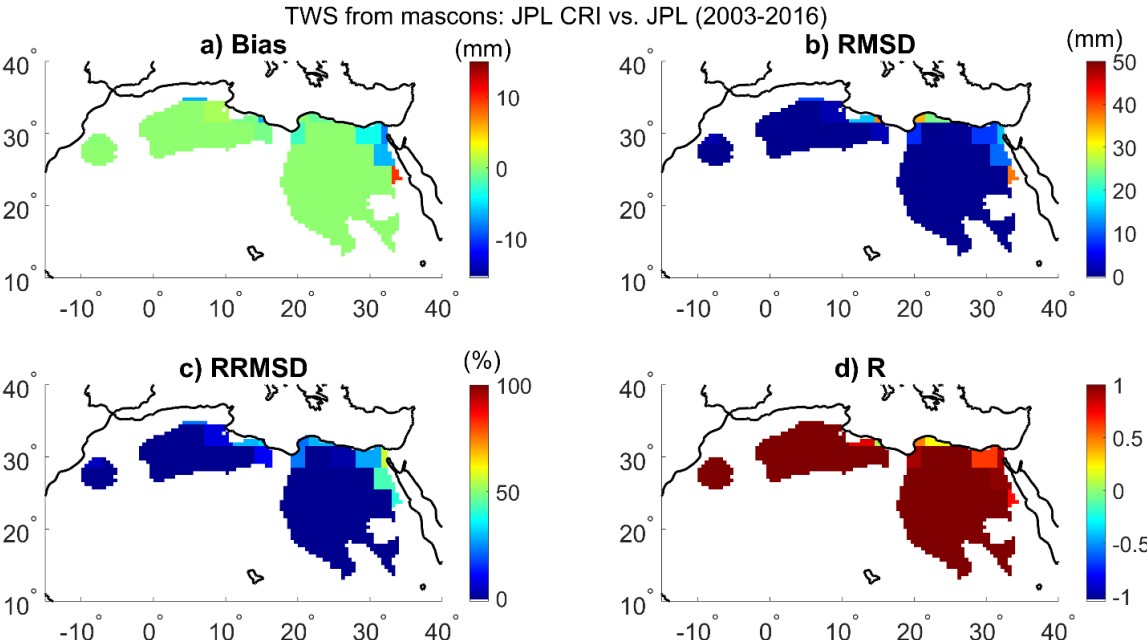

**Figure A3.** Maps of bias (**a**); RMSD (**b**) RRMSD (**c**); R (**d**) between TWS from CSR and JPL CRI mascon solutions over 2003–2016.

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
