# Peer review of "Groundwater Storage Changes in the Major North African Transboundary Aquifer Systems during the GRACE Era (2003–2016)"

_water, doi:10.3390/w12102669_

Round 1

Reviewer 1 Report

Thank you for the opportunity to review “Groundwater changes in the major North-African Transboundary Aquifer Systems during the GRACE era (2003-2016)”.  The paper provides an analysis of GRACE data to examine groundwater changes in the TBS.  The methods – use of GRACE to calculate differences in groundwater is not new and has been used extensively.  The novelty of this work is the application to the TAS.  As the methods are not new, the need to apply the method and generate greater scientific insight from it becomes the apparent focus of the paper. The work generates a dataset that illustrates the changes in the TBS over time, but the limitation of the paper is that it does not go beyond the presentation of the dataset.  As the conclusion states “This study provides an assessment of GWS changes over three major TBA located in arid and 375 semi-arid North Africa: TAS, NWSAS and NSAS during the GRACE (2003-2016).”  The is little to no interpretation of the work, and what is provided is limited in scope and support/justification.  Part of the limitation stems from the lack of developing the research question/agenda of the paper.  There is no explicit question being addressed.  In my view a more detailed analysis of what the data imply is needed before the paper should be published.

Within the discussion I note the following issues (these are also in the annotated file):

Line 317:  While this may be true, there is no data presented which supports this.  The presented data focuses on changes with nothing speaking to the factors.

Line 327-328:  The role of abstraction is inferred but how abstraction was determined calculated is not reported.

Lines 357-373:  If the NSAS is viewed  as a fossil aquifer, an explanation of the differences between the calculated loss in this work with the abstraction loss needs to be better explained. 

The last two sentences are contradictory.  The one sentence states "As increase in pumping was observed in different locations of NSAS during the last years 368 [69,70], the former value can be considered as a lower estimate.'  However, in the next sentence it is stated that the increase in rainfall resulted in less abstraction which would have resulted in a steady-state.  I can see a possible explanation here, but the author needs to address this.

Author Response

Please find my reponses to your comments in the attached document. Thank you for your suggestions.

Reviewer 2 Report

The paper analyses recent groundwater changes in the major North-African transboundary aquifers. Different datasets are described and used to derive time variations of groundwater anomalies, rainfall and evapotranspiration.

The paper is interesting and the conclusions are rightly supported by the results, however the description of the methodology is very concise and difficult to read for me. In sections from 2.2 to 2.5 the Author should provide more details. For instance, the C20 coefficient introduced in paragraphs 2.2.1 and 2.2.2 is not defined, as well as VOD in the paragraph 2.3 at line 132.

The description of the ET evaluation (lines 125-128) need more details and some references are necessary.

A few acronyms used in the text are not defined. In particular:

Line 128 Add “(SM)” between “soil moisture” and “simulations” to define SM

Lines 317 et seq. Define GWS.

Line 377 Define SWS.

Some errors and misspelled need to be corrected:

Line 30 Write “Earth”

Line 64 Replace “ad” with “and”

Line 88-90 Please, check the correctness of the sentence

Line 144 Replace “2.2.1” with “2.2.2”

Line 150 Edit the sentence “rain gauge observations. and currently available”

Lines 188 and eq. 4 The sign in the unit of measure m3.m-3 is not a point

Line 194 Replace SM with DSM

Line 205 Add “evaluation” or “estimation” in the sentence “this model provide reliable and accurate (?) of these parameters” and replace “provide” with “provides”.

Line 207 Add “of” between “from the combination” and “satellite images”

Line 211 Add “cycle” or “behavior” between “seasonal” and “can be observed”

Line 231 Please, correct the reference to “A1 and A2”.

Line 258 Write “the trend”

Line 312 Replace “Time series” with “Variations” or “Changes”

Line 317 Erase the blank space between in the word “can”

Lines 353 and 366 the sign in the unit of measure km3.yr-1 is not a point

Line 363 Add a parenthesis before “Figure 8)”.

Line 408 Replace “A.1” with “A1”

Author Response

I thank Reviewer2 for his/her overall good opinion of this manuscript. Please find my reponse to your comments in the attached document.

Round 2

Reviewer 1 Report

I appreciate the author's detailed response to my comments.  The edits/revisions that were made and documented address my concerns.  At this point, I do not have any additional comments/concerns.